# Evaluation of the environmental polio surveillance system—Northern Region, Ghana, 2021

**Benjamin Baguune**[1,2], **Eunice Baiden Laryea**[2], **Joseph Asamoah Frimpong**[2]*, **Samuel Dapaa**[2], **Kwame Kodom Achempem**[3‡], **Ernest Kenu**[2], **Dennis Odai Laryea**[3‡]

1 Environmental Health and Sanitation Department, School of Hygiene, Tamale, Ghana, 2 Ghana Field Epidemiology and Laboratory Training Programme, School of Public Health, University of Ghana, Accra, Ghana, 3 Disease Surveillance Department, Ghana Health Service, Accra, Ghana

☯ These authors contributed equally to this work.
‡ KKA and DOL also contributed equally to this work.
* asamoah.frimpong@gmail.com

**Data Availability Statement:** Yes, all data are fully available without restriction.

**Funding:** The author(s) received no specific funding for this work.

## Abstract

### Background

Acute Flaccid Paralysis (AFP) surveillance is the gold standard in the polio eradication initiative. The environmental component of polio surveillance can detect circulating Polioviruses from sewage without relying on clinical presentation. The effectiveness of the Environmental Surveillance (ES) is crucial to global polio eradication. We assessed the usefulness and attributes of the ES system in the Northern region and determined if the system is meeting its objectives.

### Methods

We conducted a descriptive cross-sectional evaluation in the Northern region from 2019 to 2020 using the updated US Centers for Disease Control and Prevention guideline. We interviewed stakeholders, reviewed records, and observed surveillance activities from 29th March to 7th May, 2021. Quantitative data were analyzed manually as frequencies and proportions whiles thematic analysis was used for the qualitative data.

### Results

One of 48 (2.1%) samples collected tested positive for circulating vaccine-derived Poliovirus (cVDPV). The cVDPV detection triggered enhanced AFP surveillance that resulted in the identification of a case of AFP. Three rounds of polio vaccination campaigns were organized. All surveillance officers interviewed were willing to continue providing their services for the ES. Reporting form has few variables and is easy to complete. The completeness of forms was 97.9% (47/48). Samples collected were dispatched on the same day to the testing laboratory. The system's data was managed manually.

**Competing interests:** The authors have declared that no competing interests exist.

## Conclusion

The system was useful in detecting polio outbreaks. Data quality was good, the system was simple, flexible, acceptable, representative, and fairly stable. Sensitivity was high but predictive value positive was low. Timeliness in reporting was good but feedback from the national level could not be assessed. There is a need to improve on the feedback system and ensure that, the surveillance data is managed electronically.

## Introduction

Poliomyelitis is an infectious disease caused by the Poliovirus which affects mostly children below the age of 15 years [1]. The characteristics of this infectious disease range in severity from a non-specific illness to severe flaccid paralysis sometimes with permanent disability [2]. Irrespective of the presence or absence of clear clinical symptoms, Poliovirus replication is considered to continue in the para-intestinal submucosal lymphatic tissue from several weeks to a few months [3]. As a result of the Global Polio Eradication Initiative (GPEI), polio cases have decreased by over 99% since 1988 [4]. The number of cases of paralytic poliomyelitis is low, there have been no polio cases due to wild type 2 or type 3 Poliovirus since October 1999 and November 2012 respectively. This was accomplished by the wide use of live attenuated oral polio vaccine (OPV) [4]. This leaves wild poliovirus type 1 and vaccine-derived polioviruses (VDPV) as the main agents responsible for polio events, with VDPV currently outnumbering the wild types [4]. A vaccine-derived Poliovirus is a strain related to the weakened live Poliovirus contained in oral polio vaccine, if allowed to circulate in under or unimmunized populations for a long time, or replicate in immunodeficient individuals, the weakened virus can revert to a form that causes illness and paralysis.

In sub-Saharan Africa, the vast majority of global polio cases were found in Nigeria [5]. Nigeria however reported its last case of Wild Poliovirus in 2016 [5]. Africa has remained polio-free since 2016 [5]. In Ghana, a total of 54 wild polioviruses (all type 1) were recorded from 1996, when Ghana launched its polio eradication initiative, to 2008, when the last wild poliovirus was detected in the Northern region [6]. Ghana was later certified polio free in 2015 after over five years of having no reported case of poliovirus [6].

A new challenge to the polio eradication process has emerged as a public health emergency due to circulating vaccine-derived poliovirus type 2 (cVDPV2) [6]. By August 2020, 323 cases of cVDPV2 and 84 cVDPV2 positive environmental samples were reported from Africa [7]. In Ghana, the first cVDPV2 positive environmental sample was recorded in June 2019 whereas the first human case of cVDPV2 was recorded in August same year [7]. At the end of 2020, 30 cases of cVDPV2 were recorded in Ghana [7].

Continuous collection and analysis of epidemiological data to monitor the trend of Polio cases is therefore crucial in the context of the Global Polio Eradication Initiative (GPEI). The AFP surveillance system is the gold standard in monitoring Polio cases. An AFP case is defined as any child under 15 years with sudden onset of paralysis or weakness in any of the limbs or any person at any age in whom a clinician suspects Poliomyelitis [6]. AFP is a complex clinical syndrome with several different etiologies including paralytic polio caused by wild Poliovirus or circulating vaccine-derived Poliovirus, Guillian-Barre syndrome (GBS), transverse myelitis, traumatic neuritis, meningitis, encephalitis, and brain tumours [8]. AFP surveillance includes the detection and investigation of all AFP cases. It has been adopted globally as an essential strategy for monitoring the progress of the polio eradication initiative [9]. AFP surveillance is

essential in every nation to timely detect paralytic poliomyelitis due to wild Poliovirus, to respond effectively to interrupt Poliovirus transmission, and to certify the absence of wild Poliovirus circulation in countries with a polio-free status [9].

The AFP Surveillance does not include the detection of the Poliovirus in the environment. Poliovirus in the mucosal lining is excreted into the faeces and shed into the environment. Hence, it is possible to have a Poliovirus in circulation in any environment contaminated with human faeces. The environment, therefore, offers an anonymous and non-invasive approach to monitor the indication of possible Poliovirus circulation in populations at risk to complement the Acute Flaccid Paralysis (AFP) surveillance system [10]. As the GPEI moves towards achieving the goal of polio eradication, many nations have included Environmental Surveillance (ES) for Poliovirus as a component of the Polio surveillance system. A person infected with the Poliovirus irrespective of whether symptomatic or not shed large amounts of Polioviruses in the faeces for several weeks after infection [11]. The ES monitors and detects the possible transmission of Poliovirus in human populations by examining environmental specimens supposedly contaminated by human faeces [11]. Several countries in Africa such as Nigeria, Kenya, South Africa, Senegal, Cameroon, Madagascar, and Ghana perform ES to detect the circulation of Polioviruses to supplement the effort of the AFP surveillance system [12, 13].

In Ghana, ES for polio is implemented in Greater Accra, Eastern, Volta, Ashanti, Northern, Bono and Bono East. The inclusion of Northern region is important because of the historic wild Poliovirus circulation. ES for polio in the Northern region started in January 2019. It involves the collection of grab sewage/wastewater samples for designated drains in the region. The samples are transported to the Noguchi Memorial Institute for Medical Research (NMIMR) for analysis. Feedback from NMIMR is sent to the reporting unit and depending on the outcome of testing, the necessary public health actions are implemented. ES for polio is a crucial component of the global polio eradication efforts. However, this component of the polio surveillance system in the Northern region of Ghana has never been evaluated. We assessed the usefulness and attributes of the ES system in the Northern region.

## Materials and methods

### Design and setting

The evaluation was a descriptive cross-sectional design. The updated US Centers for Disease Control and Prevention (CDC) Guideline for Evaluation of Surveillance Systems [14] was adopted. A mixed method approach for data collection was used to collect primary and secondary data for analysis. The evaluation was carried out from 29th March to 7th May, 2021 in the Northern region of Ghana.

The Northern region is one of the sixteen regions of Ghana. It is located in the north of the country and is the second largest of the sixteen regions, covering an area of about 26.4 thousand square kilometres. The Northern Region is divided into 16 districts (Metropolitan assembly, Municipal assemblies and District assemblies). The region's capital is Tamale. The estimated population from the 2021 population and housing census is 2,310,939. The Region has 32 hospitals, 2 policlinics, 103 clinics and health centres and 171 Community-Based Health Planning and Services compounds. Four criteria were used by a technical assessment team to select the sewer lines and sampling sites in relation to the region: (1) whether the region is classified as high risk for Poliovirus transmission, based on existing data; (2) the presence of sewer lines that receive waste from a considerable proportion of the population in the catchment area, with a minimum amount of waste coming from other areas; (3) the absence of industrial waste in the proposed site; and (4) poor AFP performance indicators. Based on these, the Technical Assessment Team recommended two sites; Nyanshegu in the Sagnarigu Municipal

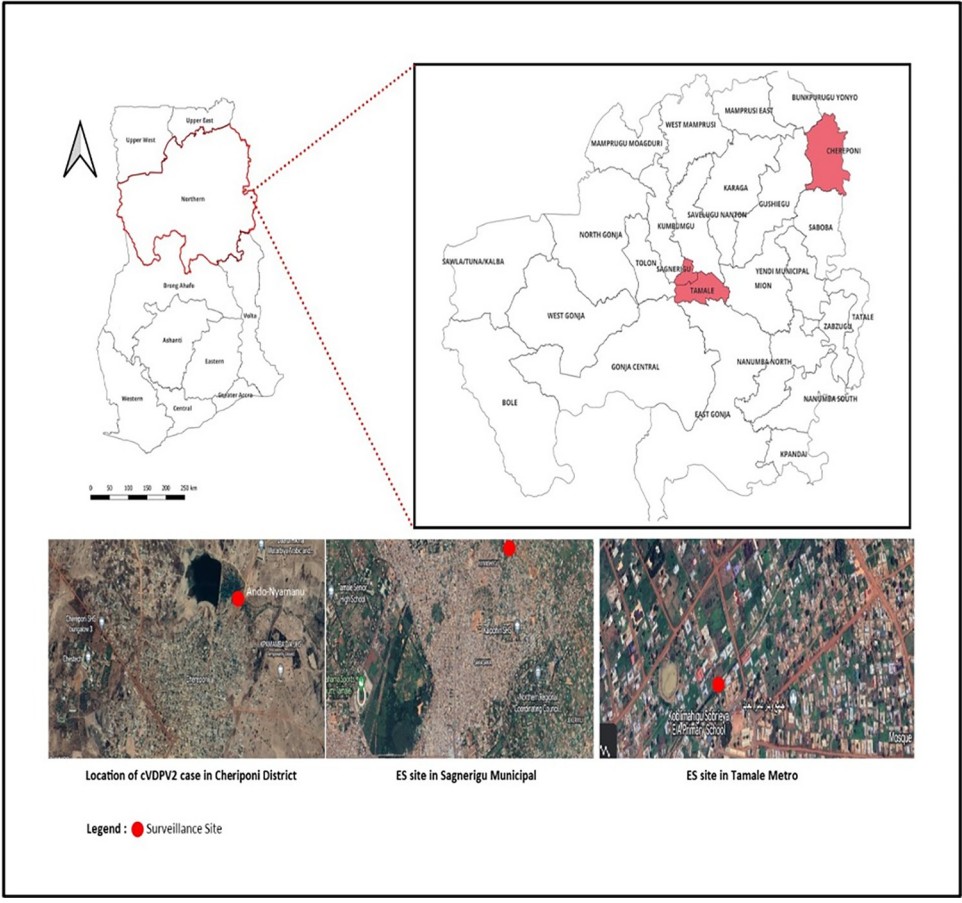

**Fig 1. Map of Northern Region showing the location of the two ES sites and the location of the cVDPV2 AFP.**

with its drainage settlements as Nyanshegu I, II & III, Sakasaka, Tishigu. Kalpohin, Kunkundan-fong, Kanbon Naayili, Ward K, Kanvilli, Tuunayilli, and Kpawumo with a total coverage population of 70,310 and Koblimahgu in the Tamale Metropolis with its drainage settlements as Old and New Koblimahgu, Old and New Jakarayih, Kpambeigu, Kukuo. Bomagu, Zozugu and Atta Esibi with a total coverage population of 32,343. The map of Northern region showing the location of the two ES sites and the location of the cVDPV2 AFP case are shown in Fig 1.

## Study participants

In addition to the observations and records reviewed, six Disease Control Officers were interviewed at the Districts and Regional level. In addition, the national environmental surveillance system's Focal Person and the Officer in-charge of the laboratory investigations at the NMIMR were also contacted through Emails and Telephone calls for certain information that could not be accessed from the Region.

## Data collection tool and method

A semi-structured questionnaire was used for data collection. Primary data was collected through face-to-face interviews with key stakeholders. The selection of key stakeholders was done through purposive sampling targeting key officials who are involved in sample collection

and transportation, documentation, data management, and use of the surveillance information. A retrospective records review from January 2019 to December 2020 was conducted to extract secondary data from the filled sewage/wastewater sample collection forms. Lastly, an observation of sewage/wastewater sample collection procedure as well as ES-related reports and guidelines were done to gather additional data. The attributes of the ES system for polio were assessed as follows:

**Simplicity.** Interviews were used to assess this attribute by reviewing the flow of information and lines of response. This includes the type and number of data collection sites, amount and type of information necessary to carry out sample analysis, methods of data transmission, number of organizations involved in receiving sample reports, type of data analysis, and methods of distributing sample information to the users.

**Flexibility.** This attribute was assessed through reviewing records and interviews, and examining the system's ability to adapt to changes in new information needs or conditions of operation.

**Data quality.** Data quality was assessed by reviewing the completeness and clarity of the completed forms for ES samples collected.

**Acceptability.** Through interviews and records review, this attribute was assessed by examining the willingness of individuals and organizations to participate in the surveillance system. It included the participation rate, the completeness of report forms and timeliness of reporting.

**Representativeness.** Through interviews and records review, the representativeness of the system was assessed by looking at the drainage coverage, distribution of data collection sites, and data collection pattern.

**Timeliness.** Timeliness was assessed through interviews and records review. It was done by comparing the time interval between samples collected, the time sample is collected, when it is dispatched to the reference laboratory and the time it takes before final laboratory results are ready.

**Stability.** Interviews were used to assess this attribute by reviewing the type of personnel involved, the ability to collect samples every month, the ability to collect, manage and provide data at a lower cost, back up when the computer system breaks down, and the number of unplanned power outages that affect the computer operating system.

**Sensitivity.** Sensitivity was assessed using the number of samples tested positive for viruses (cVDPV2, NPENT, Sabin and NPENT+Sabin) out of the total number of samples collected within the review period.

**Predictive value positive.** Predictive value positive was assessed using the number of samples tested positive for cVDPV2 out of the total viruses (cVDPV2, NPENT, Sabin and NPENT +Sabin) detected within the review period.

**Usefulness of the system.** Usefulness was assessed through interviews and reviews of outbreak reports.

## Operation of ES in the Northern Region

The operation of the ES is integrated into the existing Ghana Health Service (GHS) disease surveillance system. With regard to reporting, the same system that is being used to transport AFP stool samples and the accompanying documentation is also been used for this ES. Also, the same existing office space and AFP surveillance officers are involved in the ES. Finally, the same notification procedure and the public health actions that take place when an AFP stool tests positive, are also applied when an environmental sample also test positive. Fig 2 depicts the information flow chart of the ES in the Northern Region of Ghana.

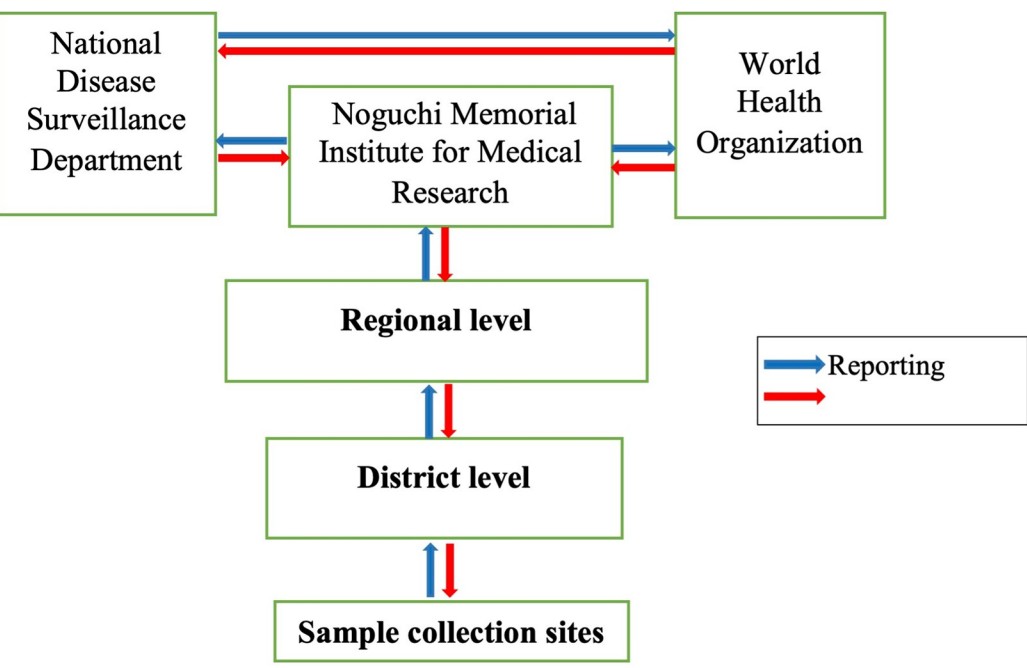

**Fig 2. ES information flow chart.**

The flow chart highlights the components and organizations of the system and how data and information is being shared between stakeholders of the system. The samples are collected from two sources; Nyanshegu in Sagnarigu Municipal and Koblimahgu in Tamale Metropolis. Grab sewage/waste water samples are collected every month from each site by trained collectors. The District Disease Control Officers in these Districts are the supervisors for the data collection exercise. One sample collection form is filled out for each sample collected. Copies of the filled forms and the samples are transported through the region to the NMIMR. The Regional Surveillance Officer also keeps a file of copies of the filled forms as well as scanned copies as backup. After laboratory investigations are done at the NMIMR, feedback is sent to the National Disease Surveillance Department, National office of the World Health Organization, Regional Health Administration and the District Health Administration.

One liter of sewage/wastewater was collected from each site every month between 6:00am and 10:00am. The early morning sample collection was done because that is the estimated peak period for the sewage/wastewater generation. The samples together with the necessary documentation is transported through the reverse cold chain system to the NMIMR. When the filled forms are sent to the region, the officer in charge inspects the forms for completeness. Data is not entered into any electronic system. The filled forms are scanned and stored as backup copies and the hard copies are kept in a file. The system's information is shared through telephone calls, Emails, and WhatsApp messages. The system is funded by the World Health Organization.

## Data analysis and presentation

Data collected at each level was reconciled with all available documents related to the ES in the study area. Quantitative data were analysed manually using summary descriptive statistics such as proportions, frequencies, and means. The ES performance indicators were analysed as follows:

**Completeness of sample collection.** This was determined by the percentage of samples collected out of the samples scheduled to be collected.

**Timeliness of sample collection.** This was determined in two ways: (1) the percentage of samples collected on the week assigned out of samples collected; and (2) the percentage of samples collected at the recommended time of day out of samples collected.

**Condition of ES sample.** This was determined by the percentage of samples that arrive in the laboratory in good condition out of samples arrived in the laboratory.

**Timeliness of ES sample shipment.** This was determined by the percentage of samples that arrive at a WHO-accredited lab < = 3 days of sample collection out of samples collected.

Thematic analysis was used to analyse the qualitative variables. All the audio recordings were transcribed while listening to the tapes and using the field notes taken. The transcripts were shared and read through thoroughly, independently, and repeatedly by three authors to ensure completeness and accuracy and also to obtain a sense of the data as a whole. Thematic analysis was carried out using usefulness and the system attributes as thematic areas. Initial codes were produced by one of the authors from list of ideas found to be interesting and relevant in the data which were later organized into meaningful groups. The generated codes were sorted out and merged to form potential themes. The initial themes were reviewed and refined into final themes taking into consideration internal homogeneity (ensuring everything in a theme is similar) and external heterogeneity (ensuring different contents in different themes). The themes were defined and named and detailed analysis conducted and written based on how they fit into the broader story of the data. Data was presented as text and graphs.

## Ethical consideration

An introductory letter was obtained from the Regional Health Directorate that was used to obtain permission from the study districts to commence the data collection process. Verbal informed consent was also obtained from the study participants before the commencement of interviews. We interviewed each participant privately and maintained the confidentiality of the data obtained. The names of participants were not mentioned anywhere in the report. The data collected was password protected. This evaluation and interviews were done as part of a routine assessment by the Ghana Health Service and Ghana Field Epidemiology and Laboratory Training Program (GFELTP) under the Ghana Public Health Act 851 [15]. Identifiable data during data collection and analysis was not accessible by the authors.

## Results

Overall, 48 sewage/wastewater samples were collected from 22nd January, 2019 to 22nd December, 2020. The mean temperature recorded for the samples collected was 23.9˚C with a standard deviation of 3.3. The ES records indicate that out of the total 48 sewage/wastewater samples collected and sent for laboratory investigations; one 1(2.1%) yielded cVDPV2, 26 (54.2%) had NPENT, 5(10.4%) had NPENT+Sabin and 1(2,1%) had Sabin 2 (Fig 3). The detection of this positive environmental isolate for cVDPV2 in June 2019, triggered an enhanced AFP surveillance in the region that also resulted in the detection of one positive AFP case of the same virus strain (cVDPV2) in August 2019 (Fig 4) in the Ando-Nyamanu community of Chereponi district, Northern region.

The ES surveillance performance indicators were also assessed. The findings show that all the indicators analyzed, performed above the world health Organization's indicator performance target levels (Table 1).

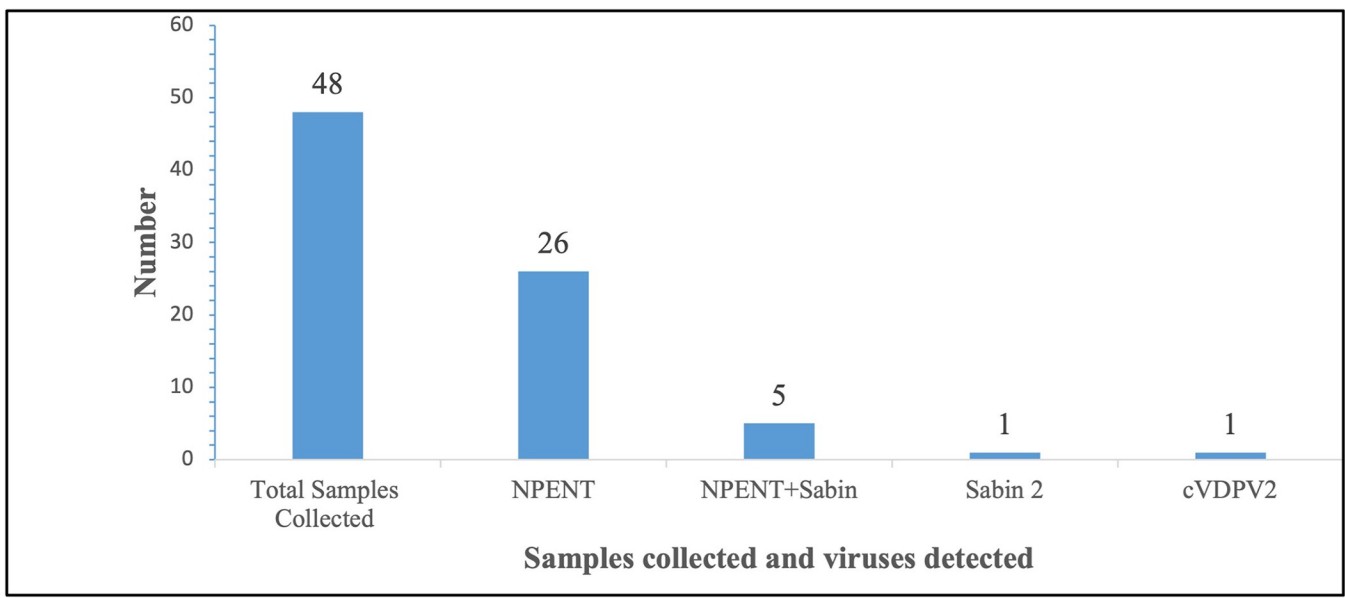

**Fig 3. Sewage/Waste water samples collected and viruses detected, Northern Region, 2019–2020.**

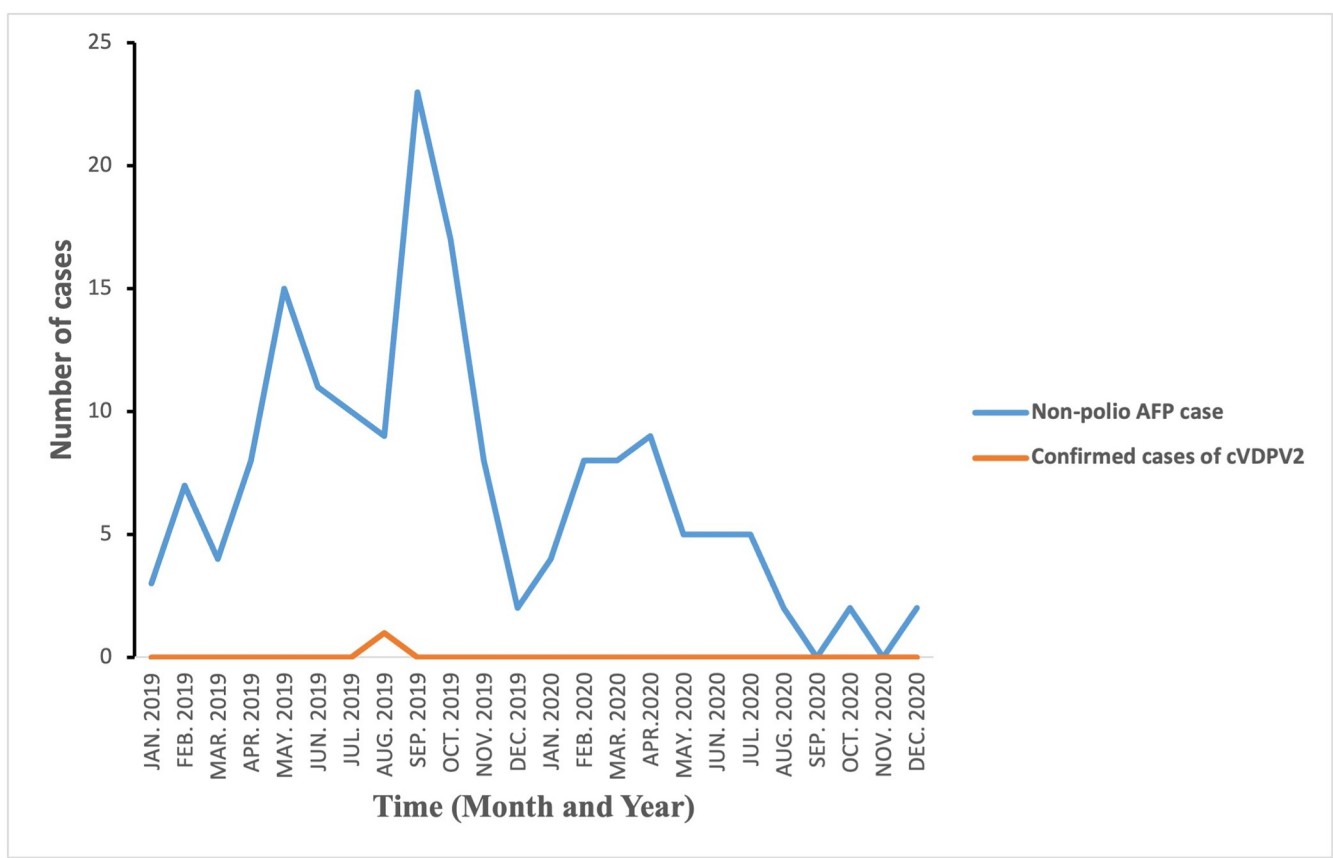

**Fig 4. AFP cases by month, Northern Region, 2019–2020.**

Table 1. ES surveillance performance, Northern Region, Ghana, 2019–2020.

| Indicator | Target | Regional Performance |
|---|---|---|
| Enterovirus detection | > = 50% | 66.7% |
| Completeness of sample collection | > = 80% | 100% |
| Timeliness of sample collection | > = 80% | 100% |
| Condition of ES sample | > = 80% | 100% |
| Timeliness of ES sample shipment | > = 80% | 100% |

## Attributes

**Simplicity.** The system's operation flow chart (Fig 2) shows that data is shared among three organizations; the GHS, NMIMR, and the World Health Organization with only two reporting sites and few surveillance officers operating it. The surveillance officers do not need any structured training, but an orientation to carry out their duties. All the 6 (100%) respondents knew the purpose of the surveillance system. Sample collection is routine and does not demand rigorous identification of cases. A one-page reporting form that contains few variables is easily filled by data collectors. Within the two years period that this evaluation was conducted, all monthly samples were collected, all forms filled and dispatched on the same day of sample collection to NMIMR, laboratory investigations were done on all the samples submitted and feedback was given to the required authorities. For data analysis, simple descriptive statistics are performed.

Some views from respondents:

*"This is not different from the surveillance work we have been doing and as such we do not need any formal training on the environmental surveillance"*

*"The environmental surveillance system help to know if there are polio viruses in the human population but are not noticed through the AFP surveillance"*

*"The system is very simple, it is just once every month that we go for data collection and the forms we use are very simple and easy to fill………. Since we started, every time we collect the samples, we send them to Accra on the same day"*

**Flexibility.** Data collected shows that, the system is integrated into the integrated disease surveillance and response (IDSR) system already in existence in Ghana. The ES system is also implemented through the same IDSR structures, surveillance officers, and reporting systems. Since already existing surveillance officers are operating the system, there was no need for any extensive training for the surveillance officers operating the system. Additionally, the system collects data on a few variables that are managed manually. One respondent said this:

*"The environmental surveillance is not a stand-alone system, it is part of the general surveillance system. We are the same officers involved so there is no rigidity in its operation"*

**Data quality.** In all, 48 forms were filled within the period for the evaluation. Out of this, 97.9% (47/48) had all the required cells completely filled and very clear to read. The data captured on the forms filed at both the district and regional levels were consistent in content and readability. With regards to data validation, data collection forms are normally inspected at the time of receiving to be sure of their completeness.

**Acceptability.** Responses generated from the interviews showed that, all (100%) the officers interviewed carried out their assigned duties without any failure and were also willing to

continue with the operation of the surveillance system despite their busy schedules. Data collectors at the surveillance sites were also willing to collect samples and fill out data collection forms every month. This was evidenced by the records reviewed which showed a 100% sample collection rate and a 97.9% form completion rate. Also, all (100%) samples were collected in the early mornings between 6:00 am and 10:00 am with the right sample collection containers and the standard ES data collection forms. All (100%) samples collected were dispatched to the NMIMR on the same day of collection. However, despite their willingness to continue with the ES activities, the data collectors suggested that more data collectors should be added to support the only two data collectors involved in ES activities.

Some views from respondents:

*". . .., just as I said, this is part of our work and we do not need additional pay to do it"*

*"If for two years now, we have been able to collect all the monthly samples, it means that we are more than ready to continue with the samples collection"*

*"The only challenge we have is the few data collectors. We wish the number should be more so if one is not there, others will stand in for the work"*

**Representativeness.** Sewage/wastewater from every section in the surveillance area is drained all year-round into the large open drainage channels within which the samples are collected. Also, through records review, it was found that, throughout the two years, samples were collected every month from the two surveillance sites and sample adequacy was consistent. A participant has this to say:

*". . ., I have never seen this gutter dried up before, even in the dry season"*

**Timeliness.** All (100%) samples were collected within the stipulated time of 4 weeks interval. It was observed that sample collection, form filling, and packaging take less than 60 minutes to complete. Samples collected were transported immediately to the Regional Surveillance office and subsequently dispatched to the national reference laboratory on the same day. It was also revealed that all (100%) samples received at the reference laboratory were processed and feedback was sent within two weeks to the national surveillance unit of the GHS as recommended.

**Stability.** There are dedicated surveillance officers at all levels (data collection sites, district level, regional level, and the national level) who are employed staff of the GHS. These officers are Government paid salary workers who indicated their willingness to work even in the absence of any extra pay. Because these staff members are already involved in AFP surveillance, they have the expected expertise to manage the operations of the ES. The system was fully operated throughout the two years without interruption from any source (personnel, electricity, or funding). Additionally, the drainage systems have good yields throughout the year. However, the system is not funded through the normal surveillance funding system. These were some views from some respondents:

*"Any time we go there, we get waste water to collect"*

*"Sample collection is not a problem at all, there is always enough waste water to fetch at any collection time"*

*"We are using our own internally generated funds (IGF) to support the system. The NMIMR provides all the logistics support, laboratory reagents, consumables, and funding for the operation of the ES through our IGF"*

**Sensitivity.** Out of the 48 samples collected, 33 yielded viruses (cVDPV2, NPENT, Sabin and NPENT+Sabin), indicating the system's sensitivity as 33/48 (68.8%).

**Predictive value positive.** Out of the 33 samples that yielded viruses (cVDPV2, NPENT, Sabin and NPENT+Sabin), only one (1) tested positive for cVDPV2, indicating the system's predictive value positive as 1/33 (3.0%).

## Usefulness

The system's data has been used to cause public health actions. It was revealed that, after the positive environmental isolate of cVDPV2 in June 2019, there was an enhancement of AFP surveillance. Surveillance Officers in the region were quickly alerted to heighten active surveillance for AFP cases and pay strict attention to completeness and timeliness of reporting. Additionally, the surveillance Officers interviewed and the records reviewed indicated that the enhancement of the AFP surveillance activities, led to the detection of the AFP case in August 2019 in the region because of which polio vaccination exercises were organized in the region.

*"After the detection of the cVDPV2 from the samples we sent, we were all alerted to enhance our AFP surveillance activities. We increase our supervision and also alerted our community health volunteers to be on the watch.., we belief this resulted in the detection of the AFP case a month later in the region"*

## Discussion

The ES provides a system for routine surveillance of Poliovirus circulation independent of clinical disease. A study in Pakistan showed that, the ES is faster in detecting polio outbreaks than the AFP surveillance system [16]. This study also found that, the first cVDPV2 polio outbreak in the study area with preceding circulation detected by both ES and AFP surveillance, the ES detected the circulation sooner. The ES detected the cVDPV2 virus circulation more than two months before detection by AFP surveillance. This is similar to the Pakistan study where the ES was found to be most effective in detecting the circulation of Poliovirus from provinces conducting ES, with detecting about three months sooner than the AFP surveillance [16]. In the same study, it was revealed that transmission during low season and reintroductions were detected by the ES in the absence of clinical cases [16].

This evaluation also found that the ES findings were very useful in Ghana to guide programmatic public health responses to Polioviruses. The advantage offered by the ES in the Northern Region was used to characterize Poliovirus transmission and to appropriately focus response efforts such as supplemental immunization campaigns. For example, after the ES isolation of cVDPV2 in Koblimahgu surveillance sites in the Tamale Metropolis and the subsequence detection of a polio case in the Cheriponi District, public health officials rolled out three rounds of successful vaccination campaigns. Using findings from ES to implement control activities has also been demonstrated in several studies such as a study conducted in Pakistan, when eight OPV immunization campaigns among children aged <5 years were used to interrupt WPV transmission as a result of Poliovirus reported from the ES [16]. The response interventions in the Northern Region contributed to the interruption of cVDPV2 transmission in the area. Furthermore, the positive isolate from the environmental sample also triggered the

conduct of a retroactive case search in the catchment area. During this house-to-house active case search, community members were sensitized on reporting AFP cases. This finding is also consistent with a study conducted in Nigeria to determine the contribution of ES toward the Interruption of Poliovirus Transmission [17]. This describes another dimension of public health intervention triggered by ES findings that helped to strengthen the AFP reporting system.

A public health surveillance system evaluation should emphasize those attributes that are most important for the objectives of the system [14]. The study assessed the attributes of a surveillance system and found that the ES in the Northern Region met most of the attributes of a good surveillance system. The system was found to be simple because data collection is routinely done with only one reporting form which contains few variables and as such easily filled by data collectors without any rigorous procedure. Additionally, only two reporting sites with few Surveillance Officers and Organizations are involved, making data sharing in the system very simple. These findings are similar to the study conducted in Bikita District, Masvingo Province [18] where the AFP system evaluated was found to be simple because of its usage of simple data tools and few reporting systems. However, these study findings are in contrast to a study in Zimbabwe, where a good number of respondents found it difficult to fill out the case investigation forms [19].

The Surveillance Officers attested to the ease of operation of the system, which is well structured, flexible to changes, and stable in its operations. The flexibility of the system has allowed the system to be integrated into the integrated disease surveillance and response system by using the same structures, surveillance officers, and reporting systems. The system has fully operated within the period of evaluation because of continuous drainage flow and dedicated surveillance officers with the requisite expertise to manage the system. Dedicated surveillance officers with requisite expertise as a measure of a stable surveillance system were also demonstrated in a study conducted in Bikita District, Masvingo Province [18]. One factor that threatens stability however is the source of funding. The NMIMR is the major funder, which provides logistics support for the transportation of samples to the reference laboratory, and the provision of some laboratory reagents and consumables. The NMIMR funds the ES from its IGF which may not be sustainable in times of insufficient IGF flow. In a similar study in Nigeria, it was also found that the majority of funding for the ES was also from partner agencies [20].

It is important to monitor data quality and thus ensure that the collected data are meaningful, so they meet the objectives of local, national, and international surveillance systems. The quality of the initial data may determine the data quality at all stages of the reporting process. Monitoring data quality also helps to improve data analysis and interpretation in public health reports at all levels. In terms of data quality, the completeness, readability, and timeliness of reporting were generally very good throughout the review period. With the exception of one form each, all the 48 forms filled within the two years period were completely filled and clear for reading. Meeting timelines with respect to data collection and transportation was 100%. Similar findings of high-quality surveillance system were seen in Oyo state [20] but contrary to findings in other similar studies, wherein some cases, the reporting was not complete and not timely [19] due to bad roads and lack of dedicated staff to transport samples to the laboratory on time. Even though data transmission to the national level was timely, we could not assess feedback due to lack of data. This can affect the future acceptability and quality of the ES operations if not immediately addressed.

The study found that the ES was generally acceptable to the surveillance officers. All the respondents were willing to continue with the ES. There was 100% execution of scheduled sample collection, transportation, and laboratory investigations by the Surveillance Officers. This finding is in line with several similar studies [18, 20]. This is probably because of the

simplicity and ease of operation of the system and also, the intensification of support by the NMIMR and the GHS and personal commitment by officers towards the eradication of polio. However, despite their willingness to continue with the current system, the sample collectors suggested that more sample collectors should be orientated to support sample collection at the surveillance sites. The system was also represented. This was because Sewage/wastewater from every section in the surveillance area is drained all year-round into the large open drainage channels within which the samples were collected. Sample adequacy also met the WHO standard and was collected every month. The system was also found to be sensitive in detecting viruses but the predictive value positive was low. This is similar to a study finding in the Oyo state [20].

### Limitations of the study

An area of deficiency observed was that, the data obtained lack information that could be used to quantify the timeliness of feedback. Additionally, the evaluation was limited by its size as more staff at all levels (District, Region National) could have also provided useful information, but the depth of the interviews yielded clear issues that are useful for the ES and GHS at large.

### Conclusion

The ES system in the Northern region of Ghana met its objective of detecting circulating Poliovirus that aided the detection of a polio case. The ES is useful in generating data for making important public health decisions including the implementation of disease control activities such as mass polio vaccination campaigns. Regarding the attributes, data quality was good, the system was simple, flexible, acceptable, representative and fairly stable. Timeliness in reporting was good but feedback from the national level was poor.

### Supporting information

**S1 File. Suspected and confirmed cases.**
(XLSX)

**S2 File. Characteristics of samples collected.**
(XLSX)

**S3 File. Questionnaire.**
(DOCX)

**S4 File. Sewage samples collected.**
(XLSX)

### Acknowledgments

The authors would like to acknowledge with sincere gratitude, the sample collectors and other surveillance staff interviewed at the Districts, Regional, and National Surveillance Departments for their support and cooperation in the data collection process. We also acknowledge the staff of the Ghana Field Epidemiology and Laboratory Training Programme for their guidance and direction throughout the course of the evaluation.

### Author Contributions

**Conceptualization:** Benjamin Baguune, Eunice Baiden Laryea.

**Data curation:** Kwame Kodom Achempem, Dennis Odai Laryea.

**Formal analysis:** Benjamin Baguune, Eunice Baiden Laryea.

**Methodology:** Ernest Kenu.

**Supervision:** Eunice Baiden Laryea, Joseph Asamoah Frimpong, Kwame Kodom Achempem.

**Writing – original draft:** Benjamin Baguune.

**Writing – review & editing:** Benjamin Baguune, Eunice Baiden Laryea, Joseph Asamoah Frimpong, Samuel Dapaa, Ernest Kenu, Dennis Odai Laryea.

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
