## [Decision Letter · Decision Letter 0]

20 Apr 2023

PONE-D-22-27652Evaluation of the Environmental Polio Surveillance System - Northern Region, Ghana, 2021PLOS ONE

Dear Mr Joseph Asamoah Frimpong,

Thank you for submitting your manuscript to PLOS ONE. After careful consideration, we feel that it has merit but does not fully meet PLOS ONE’s publication criteria as it currently stands. Therefore, we invite you to submit a revised version of the manuscript that addresses the points raised during the review process.

We look forward to receiving your revised manuscript.

Kind regards,

Enoch Aninagyei, PhD

Academic Editor

PLOS ONE

Journal Requirements:

2. We note that Figure 1 in your submission contain copyrighted images. All PLOS content is published under the Creative Commons Attribution License (CC BY 4.0), which means that the manuscript, images, and Supporting Information files will be freely available online, and any third party is permitted to access, download, copy, distribute, and use these materials in any way, even commercially, with proper attribution. For more information, see our copyright guidelines: http://journals.plos.org/plosone/s/licenses-and-copyright.

Reviewers' comments:

Reviewer's Responses to Questions

**Comments to the Author**

1. Is the manuscript technically sound, and do the data support the conclusions?

Reviewer #1: Yes

Reviewer #2: Yes

Reviewer #3: Partly

2. Has the statistical analysis been performed appropriately and rigorously? 

Reviewer #1: N/A

Reviewer #2: N/A

Reviewer #3: No

3. Have the authors made all data underlying the findings in their manuscript fully available?

Reviewer #1: Yes

Reviewer #2: Yes

Reviewer #3: Yes

4. Is the manuscript presented in an intelligible fashion and written in standard English?

Reviewer #1: Yes

Reviewer #2: Yes

Reviewer #3: No

5. Review Comments to the Author

Reviewer #1: The manuscript is well written though a few typo and grammatical errors have bee indicated in the attached document. Other technical issues have also bee raised in the document that will require the authors to address them.

Good job

Reviewer #2: Evaluation of the Environmental Polio Surveillance 1 System - Northern Region, Ghana, 2021

At a point in time, there was an estimated 75,000 paralysed children in Africa due to Polio. The disease remains a devastating one in children from low-income countries. The Global Polio Eradication Initiative came into being to assist affected countries devise strategies to curb the spread through continuous vaccination and strict surveillance. The use of environmental specimen to assist surveillance appears to be ahead of waiting to see symptoms in affected persons. This is the relevance of this paper.

Apart from a few comments seen below, I think this was a great attempt to assess the usefulness of environmental surveillance towards polio eradication.

55-56. Maybe add reasons for the global decrease in polio reports since 1988.

62. Neighbouring countries from the southern or northern part. I want to assume these outbreaks came from northern part of Ghana?

94. Northern region

335. What parameter are you using to determine delays. Is there a stipulated time that could be used to measure this? It would have been good if data is shown on national level delays a bit.

335-336. Is the quarterly submission of results from national or Nogushi the norm? Did you find out?

348. All year round

Reviewer #3: Summary

This is an important paper describing the environmental surveillance system in northern Ghana for poliovirus. The authors describe its justification and the role it plays to complement the gold standard acute flaccid paralysis case surveillance, alongside the quality and timeliness of the data

Major comments

The introduction has some inaccuracies that need to be corrected before publication. I have listed them below.

It would be very helpful to the reader to include a map of the Northern region showing the location of the two ES sites and the corresponding drainage settlements, and the location of the cVDPV2 AFP case.

It is not clear if the ES sites pass the WHO’s ES performance criteria (Table 5 of http://polioeradication.org/wp-content/uploads/2022/11/Field-Guidance-for-the-Implementation-of-ES-20221118-ENG.pdf) This is essential. It is important to document the detection of enterovirus and Sabin strains from ES sites but this is currently absent from the paper. The sites should be assessed whether they meet the WHO criteria.

Minor comments

Abstract: please include the precise dates of the study period

Abstract: The version of Excel seems excessive for the abstract

Abstract: ‘Feedbacks were not regular’ – can you quantify this statement. What proportion of results took longer than a month or another appropriate timeframe? Feedback to who?

Introduction:

Line56: 1988 – it would be clearer if you explained this is when the Eradication Initiative began.

Line 57 No wild type 3 Poliovirus since October 1999 in incorrect. Please correct this

Line 60: The last case of wild polio in Nigeria was in 2016 not 2014

Line 64: Please define what vaccine derived poliovirus is

Lines 81-82: Unfortunately seems the wrong choice of word. Also poliovirus isn’t in circulation in the environment (it’s detection can indicate circulation but it does not replicate in the environment).

Line 94: Northern region. Where else in the country is ES implemented? Why is the Northern region important? Because of historic WPV circulation?

Methods

Line 116 – how were the site locations selected?

Line 117 – what methods were used to assess the location of the drainage settlements? Was the sewage network mapped?

Line 131 – It would be helpful to include the questionnaire in the appendix

Line 156: What do you mean by ‘results are sent to the region’. Who in the region? Are they also sent to national representatives at the same time?

Line 168: ‘Stool turns positive’ I think you mean tests positive. A stool doesn’t turn positive

Results:

Line 227 – what are the other priority diseases?

Line 255 – delayed by how long?

Figure 3: Positive for what – cVDPV2? It would be interesting to show the number positive for Sabin 1,2,3 and non-polio enterovirus as well

Figure 4 – Suspected – do you mean non-polio AFP? Confirmed, confirmed for what? cVDPV2? It would be helpful to show the timings of the vaccination campaigns.

Discussion:

Line 277 – ES is faster – do you mean in this setting or more generally? If the latter please provide a reference.

Line 279 – On average, this is an average over what?

6. PLOS authors have the option to publish the peer review history of their article (what does this mean?). If published, this will include your full peer review and any attached files.

Reviewer #1: No

Reviewer #2: No

Reviewer #3: No

---

## [Author Response · Author response to Decision Letter 0]

5 Jun 2023

A rebuttal letter that responds to each point raised by the academic editor and reviewer has been uploaded.

---

## [Decision Letter · Decision Letter 1]

28 Jun 2023

PONE-D-22-27652R1Evaluation of the Environmental Polio Surveillance System - Northern Region, Ghana, 2021PLOS ONE

Dear Dr. Asamoah Frimpong,

Thank you for submitting your manuscript to PLOS ONE. After careful consideration, we feel that it has merit but does not fully meet PLOS ONE’s publication criteria as it currently stands. Therefore, we invite you to submit a revised version of the manuscript that addresses the points raised during the review process.

We look forward to receiving your revised manuscript.

Kind regards,

Enoch Aninagyei, PhD

Academic Editor

PLOS ONE

Additional Editor Comments:

The Reviewer of the revised version actually rejected the manuscript, citing the following reasons. However, I want to give you the opportunity to address them for consideration.

This publication is a descriptive summary of findings linked to an evaluation of poliovirus wastewater surveillance in Northern Ghana from 2019 to 2020. Below are the rationale for the recommendation to reject this publication:

1)The authors state that they follow the CDC guidelines for evaluating a surveillance system but they have excluded some critical assessments such as sensitivity and predictive value positive.

2) The population of the area under surveillance is 2.3million yet the sampling is done for a population of about 70,000 people, short of the 100,000 to 300,000 recommended for global polio surveillance. It is unclear how this sampling location was selected. Is this sample size sufficient for ongoing surveillance and monitoring?

3) For the subjective aspects of the evaluation, little data was shared regarding staff surveys/interviews regarding their feedback e.g. no summary of responses to specific questions though a data collection tool was mentioned.

4) Other factors to consider in evaluating wastewater surveillance particularly where there is a low risk of wild poliovirus or vdpv include the detection of Sabin strains from OPV use (i.e. correlation with immunization days) and non-polio enteroviruses (NPEV) used as proxy indicators to regularly monitor, validate, and compare site sensitivity. These indicators were not assessed.

Reviewers' comments:

Reviewer's Responses to Questions

**Comments to the Author**

1. If the authors have adequately addressed your comments raised in a previous round of review and you feel that this manuscript is now acceptable for publication, you may indicate that here to bypass the “Comments to the Author” section, enter your conflict of interest statement in the “Confidential to Editor” section, and submit your "Accept" recommendation.

Reviewer #4: (No Response)

2. Is the manuscript technically sound, and do the data support the conclusions?

Reviewer #4: No

3. Has the statistical analysis been performed appropriately and rigorously? 

Reviewer #4: No

4. Have the authors made all data underlying the findings in their manuscript fully available?

Reviewer #4: No

5. Is the manuscript presented in an intelligible fashion and written in standard English?

Reviewer #4: Yes

6. Review Comments to the Author

Reviewer #4: This publication is a descriptive summary of findings linked to an evaluation of poliovirus wastewater surveillance in Northern Ghana from 2019 to 2020. Below are the rationale for the recommendation to reject this publication:

1)The authors state that they follow the CDC guidelines for evaluating a surveillance system but they have excluded some critical assessments such as sensitivity and predictive value positive.

2) The population of the area under surveillance is 2.3million yet the sampling is done for a population of about 70,000 people, short of the 100,000 to 300,000 recommended for global polio surveillance. It is unclear how this sampling location was selected. Is this sample size sufficient for ongoing surveillance and monitoring?

3) For the subjective aspects of the evaluation, little data was shared regarding staff surveys/interviews regarding their feedback e.g. no summary of responses to specific questions though a data collection tool was mentioned.

4) Other factors to consider in evaluating wastewater surveillance particularly where there is a low risk of wild poliovirus or vdpv include the detection of Sabin strains from OPV use (i.e. correlation with immunization days) and non-polio enteroviruses (NPEV) used as proxy indicators to regularly monitor, validate, and compare site sensitivity. These indicators were not assessed.

7. PLOS authors have the option to publish the peer review history of their article (what does this mean?). If published, this will include your full peer review and any attached files.

Reviewer #4: No

---

## [Author Response · Author response to Decision Letter 1]

23 Aug 2023

Comment 1:The authors state that they follow the CDC guidelines for evaluating a surveillance system but they have excluded some critical assessments such as sensitivity and predictive value positive.

Response: We are grateful for the comment. We have revised the manuscript to provide information on sensitivity and predictive value positive

Details: Page 2, lines 44-45

 Page 8, lines 174-178

 Page 14, lines 332-336

 Page 18, lines 422-423

Comment 2: The population of the area under surveillance is 2.3million yet the sampling is done for a population of about 70,000 people, short of the 100,000 to 300,000 recommended for global polio surveillance. It is unclear how this sampling location was selected. Is this sample size sufficient for ongoing surveillance and monitoring?

Response: We are grateful for the comment. We initially only considered communities around the sample collection sites. However, other communities at the upper part of the sample collection sites feed into the drain and sewer lines but were not classified as drainage settlement. These communities have been added making the total population to be 102,653 (70,310 for Nyanshegu and 32,343 for Koblimahgu).

Details: Page 6, lines 128-133

In addition to the population (102,653), the sampling location was selected based on four criteria

Details: Page 6, lines 122-126

Comment 3: For the subjective aspects of the evaluation, little data was shared regarding staff surveys/interviews regarding their feedback e.g. no summary of responses to specific questions though a data collection tool was mentioned.

Response: We are grateful for the comment. We have revised the manuscript to include some direct quotations from respondents 

Details: Page 12 - 15, lines 270-349

Comment 4: Other factors to consider in evaluating wastewater surveillance particularly where there is a low risk of wild poliovirus or vdpv include the detection of Sabin strains from OPV use (i.e. correlation with immunisation days) and non-polio enteroviruses (NPEV) used as proxy indicators to regularly monitor, validate, and compare site sensitivity. These indicators were not assessed.

Response: We are grateful for the comment. 

The manuscript has been revised to include results on NPENT, Sabin and NPENT+Sabin viruses detected and were used in calculating the sensitivity.

Details: 

Page 14, lines 332-336

Page 10 - 11, lines 244 - 246

Figure 3 has been updated to reflect this

---

## [Editor Report · Decision Letter 2]

6 Sep 2023

PONE-D-22-27652R2Evaluation of the Environmental Polio Surveillance System - Northern Region, Ghana, 2021PLOS ONE

Dear Dr. Frimpong,

Thank you for submitting your manuscript to PLOS ONE. After careful consideration, we feel that it has merit but does not fully meet PLOS ONE’s publication criteria as it currently stands. Therefore, we invite you to submit a revised version of the manuscript that addresses the points raised during the review process.

ACADEMIC EDITOR:One of the reviewers made these comments. However, I did not see these revisions. Kindly respond to them for consideration. This publication is a descriptive summary of findings linked to an evaluation of poliovirus wastewater surveillance in Northern Ghana from 2019 to 2020. Below are the rationale for the recommendation to reject this publication:

1)The authors state that they follow the CDC guidelines for evaluating a surveillance system but they have excluded some critical assessments such as sensitivity and predictive value positive.

2) The population of the area under surveillance is 2.3million yet the sampling is done for a population of about 70,000 people, short of the 100,000 to 300,000 recommended for global polio surveillance. It is unclear how this sampling location was selected. Is this sample size sufficient for ongoing surveillance and monitoring?

3) For the subjective aspects of the evaluation, little data was shared regarding staff surveys/interviews regarding their feedback e.g. no summary of responses to specific questions though a data collection tool was mentioned.

4) Other factors to consider in evaluating wastewater surveillance particularly where there is a low risk of wild poliovirus or vdpv include the detection of Sabin strains from OPV use (i.e. correlation with immunization days) and non-polio enteroviruses (NPEV) used as proxy indicators to regularly monitor, validate, and compare site sensitivity. These indicators were not assessed.

We look forward to receiving your revised manuscript.

Kind regards,

Enoch Aninagyei, PhD

Academic Editor

PLOS ONE

Additional Editor Comments:

This publication is a descriptive summary of findings linked to an evaluation of poliovirus wastewater surveillance in Northern Ghana from 2019 to 2020. Below are the rationale for the recommendation to reject this publication:

1)The authors state that they follow the CDC guidelines for evaluating a surveillance system but they have excluded some critical assessments such as sensitivity and predictive value positive.

2) The population of the area under surveillance is 2.3million yet the sampling is done for a population of about 70,000 people, short of the 100,000 to 300,000 recommended for global polio surveillance. It is unclear how this sampling location was selected. Is this sample size sufficient for ongoing surveillance and monitoring?

3) For the subjective aspects of the evaluation, little data was shared regarding staff surveys/interviews regarding their feedback e.g. no summary of responses to specific questions though a data collection tool was mentioned.

4) Other factors to consider in evaluating wastewater surveillance particularly where there is a low risk of wild poliovirus or vdpv include the detection of Sabin strains from OPV use (i.e. correlation with immunization days) and non-polio enteroviruses (NPEV) used as proxy indicators to regularly monitor, validate, and compare site sensitivity. These indicators were not assessed.

---

## [Author Response · Author response to Decision Letter 2]

5 Oct 2023

1) The authors state that they follow the CDC guidelines for evaluating a surveillance system but they have excluded some critical assessments such as sensitivity and predictive value positive.

Response: 

We are grateful for the comment. We have revised the manuscript to provide information on sensitivity and predictive value positive

Details: Page 2, lines 44-45

 Page 8, lines 174-178

 Page 14, lines 332-336

 Page 18, lines 422-423

2) The population of the area under surveillance is 2.3million yet the sampling is done for a population of about 70,000 people, short of the 100,000 to 300,000 recommended for global polio surveillance. It is unclear how this sampling location was selected. Is this sample size sufficient for ongoing surveillance and monitoring?

Response:

We are grateful for the comment. We initially only considered communities around the sample collection sites. However, other communities at the upper part of the sample collection sites feed into the drain and sewer lines but were not classified as drainage settlement. These communities have been added making the total population to be 102,653 (70,310 for Nyanshegu and 32,343 for Koblimahgu).

Details: Page 6, lines 128-133

In addition to the population (102,653), the sampling location was selected based on four criteria

Details: Page 6, lines 122-126

3) For the subjective aspects of the evaluation, little data was shared regarding staff surveys/interviews regarding their feedback e.g. no summary of responses to specific questions though a data collection tool was mentioned.

Response:

We are grateful for the comment. We have revised the manuscript to include some direct quotations from respondents 

Details: Page 12 - 15, lines 270-349

4) Other factors to consider in evaluating wastewater surveillance particularly where there is a low risk of wild poliovirus or vdpv include the detection of Sabin strains from OPV use (i.e. correlation with immunization days) and non-polio enteroviruses (NPEV) used as proxy indicators to regularly monitor, validate, and compare site sensitivity. These indicators were not assessed

Response:

We are grateful for the comment. 

The manuscript has been revised to include results on NPENT, Sabin and NPENT+Sabin viruses detected and were used in calculating the sensivity.

Details: 

Page 14, lines 332-336

Page 10 - 11, lines 244 - 246

Refer to Figure 3

---

## [Editor Report · Decision Letter 3]

31 Oct 2023

Evaluation of the Environmental Polio Surveillance System - Northern Region, Ghana, 2021

PONE-D-22-27652R3

Dear Dr. Asamoah Frimpong,

We’re pleased to inform you that your manuscript has been judged scientifically suitable for publication and will be formally accepted for publication once it meets all outstanding technical requirements.

Kind regards,

Enoch Aninagyei, PhD

Academic Editor

PLOS ONE
---

## [Editor Report · Acceptance letter]

13 Nov 2023

PONE-D-22-27652R3 

Evaluation of the Environmental Polio Surveillance System - Northern Region, Ghana, 2021 

Dear Dr. Frimpong:

I'm pleased to inform you that your manuscript has been deemed suitable for publication in PLOS ONE. Congratulations! Your manuscript is now with our production department. 

Kind regards, 

on behalf of

Dr Enoch Aninagyei 

Academic Editor

PLOS ONE